# Experiential Culinary, Nutrition and Food Systems Education Improves Knowledge and Confidence in Future Health Professionals

**DOI:** 10.3390/nu15183994

**Published:** 2023-09-15

**Authors:** Katherine Shafto, Natalie Vandenburgh, Qi Wang, Jenny Breen

**Affiliations:** 1Department of Medicine, Hennepin Healthcare, 701 Park Avenue, Minneapolis, MN 55415, USA; 2Department of Internal Medicine and Pediatrics, University of Minnesota Medical School, 420 Delaware Street SE, Minneapolis, MN 55455, USA; 3University of Minnesota School of Public Health, and University of Minnesota Medical School, Minneapolis, MN 55455, USA; 4Chef Ann Foundation, Boulder, CO 80301, USA; 5Clinical and Translational Science Institute, University of Minnesota, Minneapolis, MN 55455, USA; wangx890@umn.edu; 6Faculty in Culinary Nutrition, Bakken Center for Spirituality and Healing, University of Minnesota, Minneapolis, MN 55455, USA; 7Faculty, College of Food Science and Nutrition, University of Minnesota, St. Paul, MN 55108, USA

**Keywords:** social determinants of health, environment, interprofessional education, nutrition for health professionals, food skills, self-efficacy, public health

## Abstract

The food system plays a crucial role in the relationship between environmental, population and individual health. While leading healthcare and environmental organizations call for urgent action to address climate–planetary–human health crises, it is often challenging for healthcare organizations to respond at a systems level to these concerns. Additionally, there is little consensus and limited research exploring how future health professionals should be trained in order to work at both the individual and systems level to address or prevent the negative health impacts related to the current food system. The intervention of a 6-week, hands-on cooking and nutrition course for graduate health professional students which examines these intersections and equips students with clinically applicable skills was examined using matched pre- and post-course surveys and thematic analysis of reflective assignments. Results indicate improved knowledge and confidence in areas including understanding the food system, guiding patients through dietary change, working interprofessionally, and applying basic nutrition concepts to clinical practice.

## 1. Introduction

The food system, including the interconnectedness of food and humans, communities and the environment, plays a crucial role in population and individual health [1,2,3]. While leading healthcare and environmental organizations call for urgent action to address climate–planetary–human health crises worldwide, there is little consensus and limited research exploring the best and most effective way to empower future health professionals to address the intersection of the food system and human health [4].

In the United States (US), over 15,000 new food products are created annually [5]. According to a study based on data from the National Health and Nutrition Examination Survey (NHANES), approximately 60% of caloric intake of adults in the US consists of ultra-processed foods, which can be defined as “essentially industrial formulations mostly or entirely made from industrial ingredients, with little or no whole foods” [6]. Recent analysis suggests the US food supply is close to 73% ultra-processed [7]. Industrially produced, ultra-processed foods (UPF) are linked to higher rates of cardiometabolic and inflammatory diseases [8].

In a systematic review conducted in 2020, among 43 studies reviewed, 37 found dietary UPF exposure was associated with at least one adverse health outcome. Among adults, these included overweight, obesity and cardio-metabolic risks; cancer, type-2 diabetes and cardiovascular diseases; irritable bowel syndrome, depression and frailty conditions; and all-cause mortality. Among children and adolescents, these included cardio-metabolic risks and asthma. No study reported an association between UPF and beneficial health outcomes [9]. In contrast, a growing number of studies support the power of cooking and home-preparation of food to improve dietary quality and chronic disease self-management [10,11].

Additionally, economic inequality is tied to disparities in food and land access, leading to disproportionate rates of chronic conditions among marginalized communities throughout the US. According to the Center on Budget and Policy Priorities (CBPP), a non-partisan research and policy institute, adults in households with very low food security are 53% more likely to be diagnosed with a chronic condition, such as hypertension, coronary heart disease, hepatitis or stroke, than adults in food-secure households [12]. Recent studies show that food skills education and home cooking not only improve health outcomes, but also improve food security [13,14].

Health sciences curricula increasingly include education linking the social determinants of health to clinical outcomes, but often fail to explore the upstream factors specific to the food system that have direct relationships to human health conditions; or the diverse experiences that play a role in shaping identities, preferences and behaviors related to food choices, food skills and food literacy.

While specific nutrition competencies for health professional students have not been officially determined, Lepre et al. conducted a critical synthesis of publications proposing nutrition competencies for medical education internationally, and identified five common themes: (1) clinical practice; (2) health promotion and disease prevention; (3) communication; (4) working as a team; and (5) professional practice [15].

We propose that clinically relevant experiential education including hands-on cooking and eating, self reflection and cross-disciplinary engagement is needed to translate awareness of system-level problems into actionable skills for future health professionals.

## 2. Materials and Methods

### 2.1. Setting and Participants

In 2016, a chef/public health nutrition expert (JB) and an Internal Medicine/Pediatrics physician (KS) with expertise in integrative medicine developed and began teaching a food-systems-focused, hands-on culinary nutrition course (Food Matters for Health Professionals, FMHP). Initially piloted with medical students, then expanded to include any health sciences graduate students, the course offers experiential, interprofessional, clinically oriented culinary nutrition and food systems education. The 1-credit, 18 h graduate level course continues to be offered 1–2 times/semester through the Earl E. Bakken Center for Spirituality and Healing at the University of Minnesota, Twin Cities Campus. 

Quantitative data from optional, identical pre-/post-course surveys were collected starting in 2018, with the description of students’ demographics and fields of study summarized in Table 1. Students enrolled in FMHP who completed both pre- and post-course surveys were included in the data analysis (*n* = 65).

### 2.2. Course Detail

The course is taught over 6 weekly three-hour sessions, each with experiential, hands-on cooking, a didactic segment with interprofessional discussion opportunities and application to clinical care or public health spheres. The curriculum includes basic, clinically oriented nutrition concepts and their relationship to health in the context of the social, structural and commercial determinants of health and various facets of the food system. Culinary skills are integrated to enhance learning about nutrition concepts (macronutrients, vitamin/mineral composition of foods, etc.). See Table 2 for the course outline.

Students are invited to share their personal and cultural experiences with food, agriculture, cooking and community, as well as their experiences as graduate students in their respective health sciences fields, through an activity during the first class session called “The Food Story” [16]. Engaging with these stories fosters cultural humility (as explained by Kibakaya et al., “cultural humility incorporates elements of self-questioning, immersion into an individual patient’s point of view, active listening and flexibility, which all serve to confront and address personal and cultural biases or assumptions”), which is increasingly recognized as an important foundation of health equity [17] (Commentary, page 2). The Food Story activity also encourages deeper exploration of both the internal wisdom in each person, and the wisdom of traditional dietary patterns from across the globe [18,19,20]. These are important perspectives for future health professionals to consider in conversations about food values with patients and communities.

Weekly culinary experiences during class provide opportunities for students to practice foundational cooking skills, interact with relevant food ingredients, learn to cook with a range of whole foods and spices and sample delicious tastes from a diversity of flavor profiles. Additional culinary assignments outside of class complement the goal of improving food literacy around choosing, procuring, preparing and consuming nourishing food. Mindful eating practices are a recurring and foundational facet of the course’s experiential elements, allowing students to practice self-awareness, examine the relationships between food and mood and to explore resources which may be supportive to both self and patient care.

Between weekly sessions, students are expected to complete assignments involving personal reflection, mindful eating experiences, review of relevant medical and nutrition literature and an online discussion forum about a personal cooking experience. The final project involves a patient case representing common clinical conditions in which students, assigned to interprofessional groups, are asked to design a meal recommendation as well as a patient education handout. The group creates a teaching video demonstrating their meal suggestion, explaining the relevance to the patient’s condition(s) and describing the cost per serving. The group also submits a brief academic paper, with supporting literature, for the rationale of their meal suggestion pertaining to the patient’s specific condition(s).

Of note, the elements of this course have been consistent throughout the last 7 years, though the delivery shifted to virtual due to the COVID-19 pandemic, and has continued as such based on feedback from students, as well as the increased access the virtual platform allows. See Figure 1 for the teaching methods used in the course. 

### 2.3. Data Collection and Statistical Analysis

Identical pre- and post-course surveys were administered through the REDCap^®^ (Research Electronic Data Capture, version 13.1.37) system, hosted at the University of Minnesota, independent of the course evaluation process. Survey data, managed in REDCap, were de-identified and instructors were not aware of which students completed surveys. Completion of the surveys was optional for students and did not impact their grade. The study was determined to be exempt from full Institutional Review Board review.

Survey questions were developed after a pilot course indicated priority content including personal wellbeing; habits and approach to food; knowledge and skills to help a patient adapt a recipe according to dietary patterns or preferences; knowledge in relation to clinical practice; and application to clinical practice. Survey questions referenced specific topics or concepts that were addressed in the course, as well as knowledge or skills highlighted in the course objectives. By using an identical pre- and post-course survey, matched responses from students follow the same timeline relative to the intervention (the 6-week course). An appropriate five point Likert scale was created for each section of the survey. The draft survey questions were reviewed by three experts in the fields of medicine, nursing and public health, and validated for content. The complete survey questionnaire, “Food Matters for Health Professionals Survey”, is presented in the Appendix A section.

Participants’ demographics and characteristics were summarized using mean and standard deviation (SD) for continuous variables and frequency and percentage for categorical variables. Survey responses were summarized using mean and SD. Pre–post change was evaluated using a paired *t* test. Statistical analyses were performed in SAS version 9.4 (SAS Institute Inc., Cary, NC, USA). *P* values of less than 0.05 were considered statistically significant.

Qualitative data were collected from a post-course reflective assignment which asks students to respond to the following prompts: (1) Choose one or two specific lessons or themes that resonated personally with you in some way. Write a 1–2 paragraph description of how this has or will impact your life or your eating in the days ahead. (2) Write a 1–2 paragraph description of how you can use what you learned from Food Matters in your professional context. De-identified, collated reflective assignments from six course cohorts (*n* = 69) were reviewed for frequency of codes derived through inductive coding.

## 3. Results

Changes in attitudes and knowledge were statistically significant in all areas, including personal well-being; habits and approach to food; knowledge and ability related to food and culinary practices; knowledge of dietary patterns; knowledge and skills to help a patient adapt a recipe according to dietary patterns or preferences; knowledge in relation to clinical practice; and application to clinical practice.

Figure 2, Figure 3, Figure 4 and Figure 5 display the changes between participants’ pre- and post-course survey responses for the following categories: “Knowledge to discuss dietary patterns/styles with a patient” (Figure 2), “Knowledge and skills to help a patient adapt recipes to a specific diet” (Figure 3), “Clinical application of specific nutrition concepts” (Figure 4) and “Practices around culinary nutrition in the clinical setting”, (Figure 5).

The greatest changes between pre- and post-course survey responses were found in the following areas: discuss different dietary patterns with a patient (Figure 2); adapt a recipe according to dietary patterns or needs (Figure 3); specific knowledge concerning the relationship of food, nutrition and clinical conditions (gut health and fats, Figure 4); recommend dietary modifications to a patient with a chronic disease; and guide a patient through steps of dietary changes and food preparation techniques (Figure 5).

Inductive coding, starting with review of 10 post-course reflections to identify 11 codes, and subsequent coding of 69 student reflections yielded the highest frequency of references to the topic of food skills, followed by translation of knowledge to patients/clients, communicating with peers/family/community about what they learned and food systems concepts. Additional codes were found less frequently. Because this was a course assignment, these responses represent experiences from all students in the cohorts examined. The percentage of reflections mentioning each theme and example comments of each identified theme are provided in Figure 6 and Table 3.

## 4. Discussion

Based on the quantitative and qualitative data gathered, experiential culinary education improved food skills and self-efficacy among survey respondents. Students reported improved knowledge of food systems, food skills, basic nutrition concepts and relationship to chronic disease conditions. Understanding and recognizing the spectrum of quality of foods and food products through a critical review of nutrition labels and ingredient lists, and having practical food skills to improve dietary quality, equips future health professionals to support patients and communities as they navigate the complexity of today’s food landscape. 

Through our data analysis from both surveys and reflective responses, we see that not only did students’ knowledge of basic nutritional and culinary concepts increase between pre- and post-course, but their ability to apply the knowledge to clinical or professional practice also increased. An increase in self-perceived abilities and self-efficacy translates into confidence in practice [21].

This study has the following limitations: The pre-/post-course survey was optional for students enrolled in the course, and only survey responses with both pre- and post-course responses were used. The course is only required for the Integrative Health and Healing Doctorate of Nursing Practice (DNP) students, so for the majority of students, the course is elective. Self-selection or a predetermined interest in the subject matter could also influence students’ enrollment in the class, as well as pre- and post-survey responses. The majority of students responding to the survey identified as female, with only a small portion indicating male gender. There was no control group, so there is no standard of comparison with which to measure knowledge or attitudes on the topic areas. Because of these factors, the results of this study may not necessarily be generalized to a larger or more diverse population of graduate health sciences students.

Despite these limitations in the quantitative data, the qualitative data reflect the experience of all students enrolled in the cohorts examined and ultimately complement the survey results. Among the strengths of this study are the length of time over which the data have been collected, the diversity of health sciences fields represented and the alignment of the findings from qualitative and quantitative data.

Given that the qualitative data show a high frequency of “food skills” and “translation of knowledge to patients/clients” themes, an area of future research could explore the impact of an experiential food systems and nutrition course as a required part of health sciences curricula.

## 5. Conclusions

Health Sciences educators, major medical journals, Congress (see 2022 House Resolution 1118 that called “on medical schools, graduate medical education programs, and other health professional training programs to provide meaningful physician and health professional education on nutrition and diet”) and the larger public are calling for improved nutrition education for future physicians and health professionals [22,23].

We suggest that sharing experiences around food with other people from diverse fields of study and/or backgrounds promotes the development of cultural humility—a key ingredient for compassionate, informed and well-rounded health professionals, as well as humans. The benefit of an interprofessional course combined with elements such as hands-on cooking, the Food Story activity and examination of various dietary patterns appears to make an impact on students’ self-reported confidence and knowledge of food preparation and nutrition information.

These findings align with the growing body of literature supporting culinary and nutrition education for health professionals, yet includes the unique elements of food systems and interprofessional education. In order for nutrition education to be meaningful and memorable (and therefore effective), we believe our research indicates that such education needs to be experiential, clinically relevant, interprofessional and systems-oriented.

## Figures and Tables

**Figure 1 nutrients-15-03994-f001:**
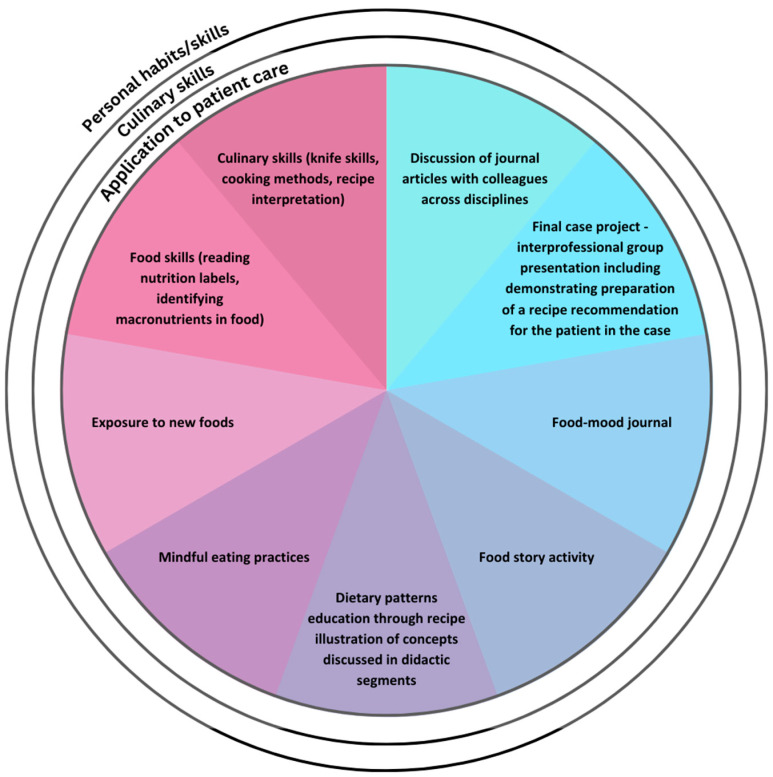
Experiential methods employed in FMHP Course targeting personal, culinary and patient-care skills.

**Figure 2 nutrients-15-03994-f002:**
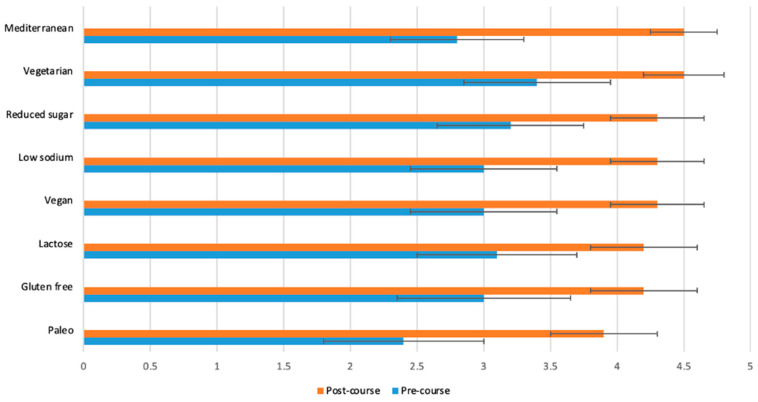
Students (*n* = 65) were asked before and after the course to rate their response to the following statement on a scale from 1 (poor) to 5 (excellent): “I have sufficient knowledge to discuss each of the following diet styles with a patient or client”. The *p*-value for the change between pre- and post-course values was <0.0001 for all variables.

**Figure 3 nutrients-15-03994-f003:**
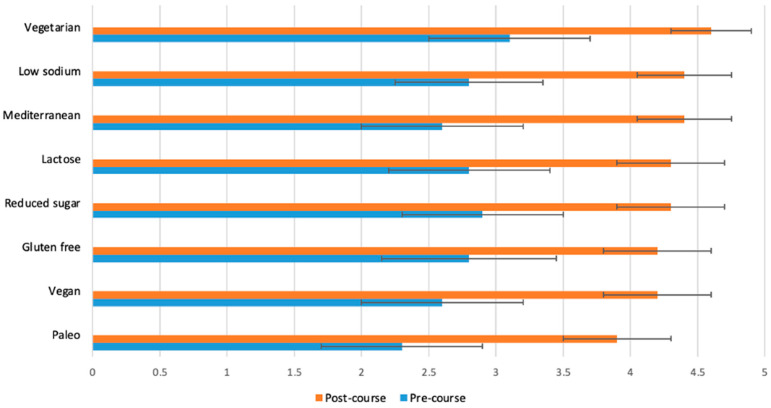
Students (*n* = 65) were asked before and after the course to rate their response to the following statement on a scale from 1 (poor) to 5 (excellent): “I have sufficient knowledge and skills to help a patient adapt a recipe to each specific diet”. The *p*-value for the change between pre- and post-course values was <0.0001 for all variables.

**Figure 4 nutrients-15-03994-f004:**
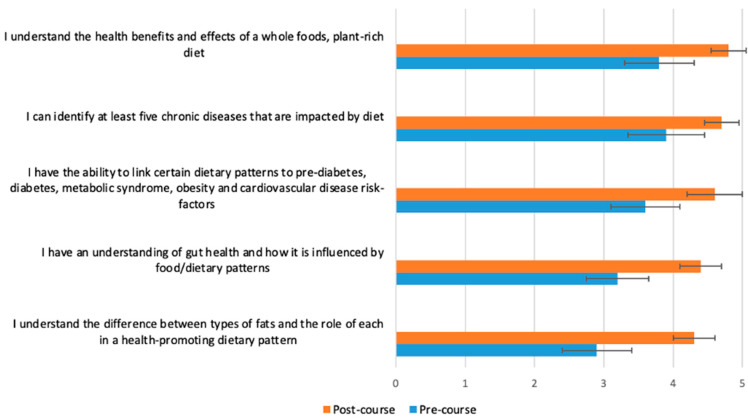
Students (*n* = 65) were asked before and after the course to rate their response to the following statement on a scale from 1 (poor) to 5 (excellent): “Please rate your level of knowledge in the following areas”. The *p*-value for the change between pre- and post-course values was <0.0001 for all variables.

**Figure 5 nutrients-15-03994-f005:**
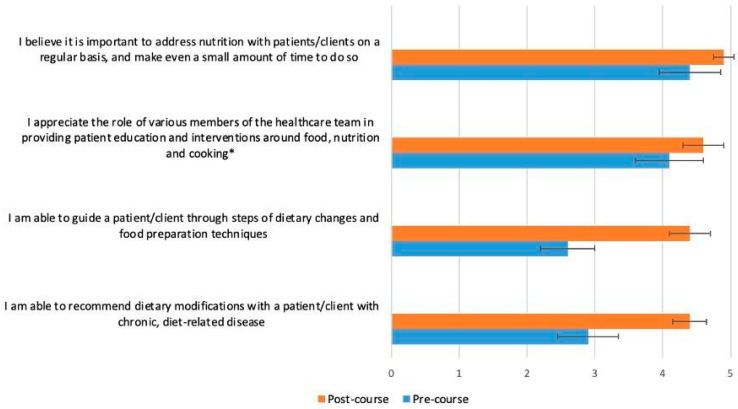
Students (*n* = 65) were asked before and after the course to rate their response to the following statement on a scale from 1 (strongly disagree) to 5 (strongly agree), with intermediate choices of 2 (disagree), 3 (neutral) and 4 (agree): “Please rate your practices around culinary nutrition in the clinical setting”. The p-value for the change between pre- and post-course values was <0.0001 for all variables, except the one marked with an asterisk (*), which had a *p*-value of 0.0005.

**Figure 6 nutrients-15-03994-f006:**
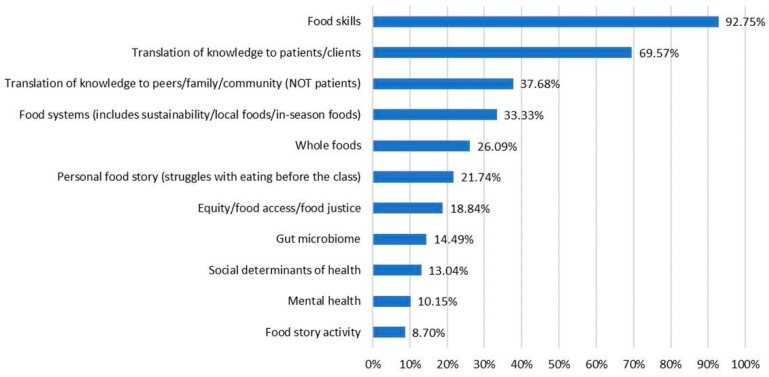
Themes from reflective assignments and percentage of assignments mentioning each theme (*n* = 69). Students were instructed to reflect on two questions: “(1) Choose one or two specific lessons or themes that resonated personally with you in some way. Write a 1–2 paragraph description of how this has or will impact your life or your eating in the days ahead. (2) Write a 1–2 paragraph description of how you can use what you learned from Food Matters in your professional context”. De-identified, collated reflective assignments from six course cohorts (*n* = 69) were examined through an inductive coding process.

**Table 1 nutrients-15-03994-t001:** Participant demographics (*n* = 65).

Variable	Characteristic		Frequency	Percent (%)
**Role**	Student		51	78.5
	Both		12	18.5
	Clinician		2	3.1
**Gender**	Female		60	92.3
	Male		5	7.7
**Program of study or clinical area**	Doctor of Nursing Practice		18	27.7
		Integrative Health and Healing	9	13.8
		Family Nurse Practitioner	4	6.2
		Specialty not specified	4	6.2
		Psychiatry	1	1.5
	Public Health		10	15.4
		Healthcare Administration	3	4.6
		Nutrition/Dietetics	3	4.6
		Community Health Promotion	1	1.5
		Epidemiology	1	1.5
		Maternal and Child Health	1	1.5
		Specialty not specified	1	1.5
	Pharmacy		9	13.8
	Integrative Health and Wellbeing Coaching		6	9.2
	Health Coaching		3	4.6
	Family Nurse Practitioner		2	3.1
	Integrative Health and Healing		2	3.1
	Nursing		2	3.1
	Women’s Health		2	3.1
	Health and Wellbeing Sciences		1	1.5
	Integrative and Functional Medicine		1	1.5
	Medical Student		1	1.5
	Nutrition		1	1.5
	Occupational Therapy		2	3.1
	Oncology		1	1.5
	Psychology		1	1.5
	Senior Citizen Education		1	1.5
	Unknown		1	1.5
	Youth Development Leadership		1	1.5
			**Mean**	**SD**
	If clinician, number of years in practice		13	9.9

**Table 2 nutrients-15-03994-t002:** Course content by week and Course Learning Objectives.

Week	Course Content
Week 1	Introductions; Food Story; culinary basics including knife skills, salad dressing formula, dry heat cooking methods, use of seasonal and local ingredients; overview of local/regional and global food systems and their relationships to health and social/structural determinants of health.
Week 2	Overview of digestive physiology; mindful eating in practice; overview of macronutrients; comparison of food plates/graphics; Western dietary pattern and chronic disease relationships; incorporating whole grains, legumes and vegetables into a variety of recipes; moist heat cooking methods.
Week 3	Quality and sources of macronutrients, dietary patterns and metabolic syndrome; case examples; functions of fat in culinary and nutritional context; soups/stews formula, nutrient-dense dessert examples.
Week 4	Gut microbiome foundations, influence of diet on gut health, dietary pattern influence on gut health/mental health/gut–brain axis; food and inflammation; identifying, preparing and incorporating fermented foods.
Week 5	Cost and time considerations, building a pantry, budgeting, using leftovers; examining literature around popular dietary patterns; the animal protein flip, food preservation techniques, build-a-bowl formula.
Week 6	Final group case presentations and peer teaching opportunity, application to professional practice, personal and group reflection.
Course Learning Objectives	
1.	Describe and assess the interconnections between environmental, cultural, economic, social, nutritional and personal issues within the food system as it relates to production, processing, purchasing, consumption and food choices.
2.	Demonstrate an understanding of the diversity of perspectives of various participants in the food system and critically evaluate information with an understanding of these various perspectives.
3.	Critically examine personal cultural perspectives and roles in the food system, both professionally and personally, and describe opportunities for public engagement in issues of personal importance.
4.	Demonstrate basic cooking skills, including troubleshooting and problem solving in the context of meal preparation.
5.	Demonstrate a working understanding of basic nutritional and food safety guidelines.
6.	Create weekly/monthly menus within a realistic food budget.
7.	Demonstrate how to adapt and create nutritionally balanced meals within a variety of constraints (e.g., using available ingredients, budget constraints, and dietary and cultural preferences, etc.).

**Table 3 nutrients-15-03994-t003:** Selected quotes from each theme identified in students’ reflective assignments. (*n* = 69).

Theme	Quote
Food skills	The cooking skills and practices made me a lot more comfortable in the kitchen and broadened my horizons on what foods to cook.
Translation of knowledge to patients or clients	While I know what foods are generally healthy, I didn’t have a good understanding of how I could help patients with their nutrition. This class gave me the opportunity to focus on just that, and I feel better equipped to discuss diet and nutrition with my future patients. One of the main lessons I took away from this course was how to meet your patients where they are at, and see what small adjustments the patient is comfortable with.
Translation of knowledge to peers, family, or community (NOT patients)	I think that this class was helpful in reaffirming my attitude towards medication, and it will allow me to be an advocate for diet/lifestyle modifications within the pharmacy community.
Food systems (includes sustainability, local foods, and in-season foods)	Due to this expansion in my understanding, I now have a greater awareness of figuring out the true source of the organic food products I am purchasing. “Was it grown locally?” “How far did it have to travel from farm to store?” These are some of the questions I now ask myself while grocery shopping.
Whole foods	My first takeaway is the emphasis on whole foods and how that benefits health.
Personal food story (struggles with eating before the class)	Another lesson that resonated with me was how oversimplified calories in/calories out is. I have struggled with my weight in the past and have always bought into the notion that calorie restriction was really the only way to lose weight. I think it really comes down to the idea that losing weight and eating/being healthy are two very different things.
Equity, food access, and food justice	I am grateful this class allowed and fostered a space to discuss inequities and inequalities in our food systems.
Gut microbiome	Secondly, our discussions and lessons on how to maintain a healthy gut were so great! It is very cool to see more and more research about how important it is to maintain, how it changes over time, and how it differs from person to person. I do notice that when I eat well and eat fermented food regularly, my gut is happier!
Social determinants of health	The class highlighted that eating healthy isn’t exactly difficult. However, there are many factors that may contribute towards an individual making poor food decisions. Stress, access to healthy grocers, financial resources, and health literacy are just a few reasons for bad eating habits. It can be very easy to recommend foods to a patient, however it can be very difficult to address all factors that may contribute to a person or family’s difficulty in eating healthy.
Mental health	Utilizing the resources for gut health, the microbiome and inflammation connection as well as the link between real food and brain health are also key elements to bring to my coaching practice when working with clients contending with inflammatory and mental health issues.
Food story activity	We have discussed mindful eating in my coursework already, but it was very helpful to remember the elements and also experience it for myself to speak to clients about my own experience around it. I am looking forward to adding the Food Mood diary and the Food Story into my coaching toolbox to use with clients.

## Data Availability

The data presented in this study are available on request from the corresponding author. The data are not publicly available due to privacy restrictions.

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
