# Peer review of "Experiential Culinary, Nutrition and Food Systems Education Improves Knowledge and Confidence in Future Health Professionals"

_nutrients, 2023, doi:10.3390/nu15183994_

Round 1
Reviewer 1 Report
This is a well written and interesting paper. It is well positioned and very clear. I just have a couple of minor suggestions.
Pg 4 line 108 - it is good form to provide the page number when you quote directly
Pg 8 line 200 Please explain the type of thematic analysis conducted - this is important e.g., open and then axial coding Corbin and Strauss (1999); who conducted the analysis etc.
Author Response
Dear Reviewer 1: Thank you for your helpful feedback. We have incorporated your suggestions as follows:
Line 108: adding page number for the direct quote in introduction
Response: New lines 134-139: Engaging with these stories fosters cultural humility (as explained by Kibakaya et al, “cultural humility incorporates elements of self-questioning, immersion into an individual patient's point of view, active listening and flexibility, which all serve to confront and address personal and cultural biases or assumptions”), which is increasingly recognized as an important foundation of health equity.[17, Commentary, page 2]
Line 200 additional details regarding the qualitative/coding data analysis.
Response: Lines 199-200: De-identified, collated reflective assignments from six course cohorts (n=69) were reviewed for frequency of codes derived through inductive coding.
and later in the Results section:
New Lines 240-244: Inductive coding, starting with review of 10 post-course reflections to identify 11 codes, and subsequent coding of 69 student reflections yielded the highest frequency of references to the topic of food skills, followed by translation of knowledge to patients/clients, communicating with peers/family/community about what they learned, and food systems concepts. Additional codes were found less frequently.
We appreciate your feedback and contributions to our manuscript.
Reviewer 2 Report
Dear authors,
The manuscript title is “Experiential culinary, nutrition and food systems education improves knowledge and confidence in future health professionals” and it aims to perform and intervention of a 6-week hands-on cooking and nutrition course to health professional students.
The topic is very original, emergent and falls within the aims and scope of the journal. It is very important in the context of climate changes, environmental and global health. The abstract is clear and complete. I would suggest the authors not to repeat in keywords words from the title. Some keywords seem too long. The background fits the theme well.
Some particular suggestions/comments will be done here:
- Line 38 – United States (US)
- Line 43 – if you used the acronym (US) before, you should use from then on
- Line 62 – I would say “improves food security” – you mean a benefit right?
- Line 143 – please add a ® in front of RedCap
- Lines 146/147 – you need to write full the meaning of IRB
- Lines 148 – 153 – If I well understand the tool you use in this survey was the questionnaire; you need to add it in supplementary material or to add on the text the full questions made; also, it is not clear how did you define the questions to be made, you need to detail a little bit more about the methodology used to the construction of the questionnaire
- Line 160 – it is not clear for me, you made the same questionnaire before and after the course, right? Then an extra qualitative analysis only after the course. Did I get it right?
- Lines 168 – 172 – these variables are a little different from the ones said before (149-152) which can cause confusion in readers
- Line 178 – I would suggest authors to write something between figures like a phrase highlighting the most important result or a brief comment because we should not add several figures followed with no text.
- Line 226 – although you have done quantitative analysis and collected same data on qualitative analysis I believe it is too ambitious to call this a mix-method study
- Lines 238 – 240 – I agree with the authors also because the “n” is very low, so this study may give same tendencies only, but important anyway
- Line 266 – for the same reason as before, I would not say “affirms” but “suggests” or “indicates”
I had not access to any supplementary material.
Congratulations! This reviewer would love to attend your course!
